# SMAAT: Scalable Manifold-Aware Adversarial Training for Large Language Models

## Abstract

Adversarial Training (AT), the method of finetuning a deep learning model with adversarially generated examples, is the most reliable form of making a model robust against future adversarial perturbations. However, AT is substantially expensive than standard training as it requires several full forward and backward passes to compute adversarial examples. In this paper, we introduce SMAAT, an efficient AT method that uses only adversarial examples generated in the last layer to finetune encoder-based large language models. The basis of our approach are the following three observations (i) the intrinsic dimensionality of the embedding space spanned by different layers of a deep model is substantially lower than the explicit dimensionality of the token embeddings; (ii) Encoder-based language models exhibit a monotonic behavior in their intrinsic dimensionality, i.e., deeper layers (closer to the output) have much lower intrinsic dimensionality than the shallow layers (closer to the input); (iii) off-manifold examples tend to persist across layers, i.e., an image of an off-manifold example generated in a shallow layer continues to remain off-manifold with respect to the embedding space of the later layers. We empirically demonstrate the effectiveness of SMAAT and show that it increases robustness by 8.6%, 15.7%, and 28.8% for BERT and 6.0%, 5.8%, and 19.0% for RoBERTa over the previous state-of-the-art results on AGNEWS, IMDB, and YELP, respectively. These improvements are achieved while maintaining comparable generalization and reducing the computational cost to approximately 1/3 to 1/4 of the GPU times required by the Projected Gradient Descent algorithm.

## 1 Introduction

Adversarial training (AT) has been shown to be the most effective approach for learning deep neural network models that are robust to perturbations to the input (Bai et al., 2021; Kurakin et al., 2017). AT is formulated as a min-max optimization problem where the outer minimization is over the parameter space and the inner-maximization searches for the worst-case input space perturbation. For deep learning models, the inner maximization is solved approximately using several iterations of the projected gradient descent method (PGD, Madry et al. (2017)). PGD is expensive as it requires forward and backward passes over the full length of the network. Thus, $P$ iterations of PGD would require $P$ additional forward and backward passes. As deep models, in particular language models, are becoming increasingly larger (Devlin et al., 2018; Brown et al., 2020), the cost of AT is becoming prohibitive (Schmidt et al., 2018), potentially leading to models being released in the wild without robustification against adversarial attacks.

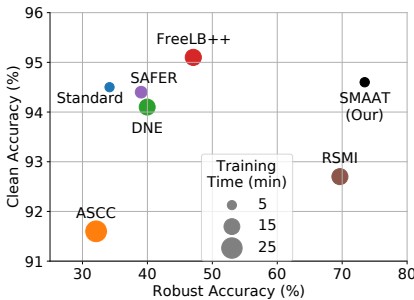

Figure 1: Evaluation on AGNEWS dataset using BERT. SMAAT achieves SOTA results on robustness and scalability while maintaining comparable clean accuracy to standard training. The size of the marker is proportional to the runtime.

Recently, a growing body of work has been focusing on making AT scalable. Specifically, several approaches have been proposed to eliminate the overhead cost of generating adversarial examples (AEs), by recycling the gradient information computed when

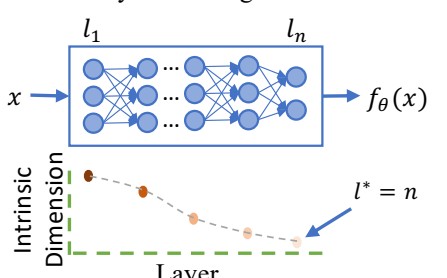 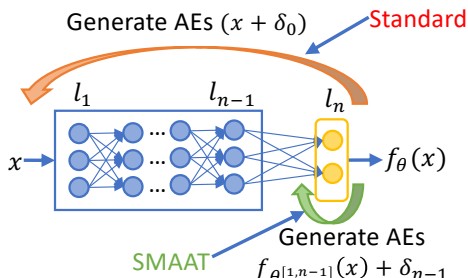

Figure 2: Overview of SMAAT. First, SMAAT searches for the layer with the highest index $l^*$ where preceding layers have monotonically decreasing intrinsic dimension. Perturbations to $l^*$ result in the shortest PGD forward-backward chains (more scalability) and the highest proportion of off-manifold AEs (more robustness by the manifold conjecture). Next, SMAAT trains the network adversarially by generating samples from $l^*$ as opposed to $l = 1$, as classically. This results in $\mathcal{O}(K(n - l^*)(max(\{d_i\}_{i=1}^{l^*-1}) - d_{l^*}))$ gain in run-time complexity per training iteration.

updating model parameters to generate the perturbations.In particular, *FreeAT* Shafahi et al. (2019) combines parameter updates and ascent steps for input perturbation. Similarly, YOPO (Zhang et al., 2019) accumulates gradients over ascent steps. Such approaches, however, suffer from the "stale gradient" problem (Dutta et al., 2018), where the samples do not guarantee to optimize neither of the min or max programs. Similarly, FreeLB (Zhu et al., 2020) proposes a better solution with updating the parameters with the accumulated gradients at every step of PGD. This however still results in additional training iterations. Further optimizations include AT with 1-step PGD, *i.e.,* fast gradient sign method (FGSM), with random initialization (Wong et al., 2020).

Instead of reducing the number of PGD steps, we propose SMAAT which achieves scalability by reducing the length of the PGD forward-backward passes. SMAAT is inspired by the following observations: (i) The latent representations used by the layers in encoder-based language models (LMs) reside on data manifolds with lower dimensionality than the overall dimension of the embedding space; (ii) AT with off-manifold examples results in better robustness while AT with on-manifold examples results in better generalization (Minh & Luu, 2022); (iii) adversarial examples are likely to leave the underlying low-dimensional data manifold Ethayarajh (2019); Shamir et al. (2021); Gilmer et al. (2018); and (iv) LMs exhibit a monotonically decreasing intrinsic dimension (ID) behavior where layers deep in the network have much lower ID than the shallow layers. Here, ID is the manifold dimensionality of the layer. SMAAT builds on these findings and identifies the layer with the lowest intrinsic dimension to generate AEs. This leads to (1) better scalability as it reduces the PGD forward-backward chains length from $n$ (number of layers) to $n - l$, where $l$ is the layer with the lowest dimensional manifold, and (2) higher robustness as the selected layer is likely to result in the highest proportion of off-manifold AEs.

We adversarially train BERT-base (Devlin et al., 2018) and RoBERTa-base (Liu et al., 2019) models on three NLP tasks, *i.e.,* AGNEWS, IMDB and YELP. SMAAT achieves state-of-the-art (SOTA) results on all tasks across two metrics: (1) run-time and (2) robustness, while maintaining comparable clean accuracy to standard training (*e.g.,* Figure 1). SMAAT enhances robustness by 8.6%, 15.7%, and 28.8% for BERT and 6.0%, 5.8%, and 19.0% for RoBERTa over the previous SOTA results on AGNEWS, IMDB, and YELP tasks, respectively. This is achieved while maintaining clean accuracy. Moreover, SMAAT requires only $\approx$1/3-1/4 of the GPU time used by PGD during training.

## 2 RELATED WORK

**Adversarial training** aims at robustifying a model against small perturbations to the input that are imperceptible to humans but can result in a wrong prediction. This is achieved by training the model to correctly classify perturbed versions of the data samples - AEs. In the text domain, AEs can be

generated at different granularity levels; (1) by adding, deleting or replacing characters (Gao et al., 2018; Ebrahimi et al., 2017; Li et al., 2018), (2) by word substitution (Ren et al., 2019b; Li et al., 2020b; Garg & Ramakrishnan, 2020; Alzantot et al., 2018) (3) by manipulation of entire sentences (Jia & Liang, 2017; Iyyer et al., 2018; Zhao et al., 2017) or (4) by perturbing the embeddings (Altinisik et al., 2022). Formally, AT is framed as a robust optimization program. Specifically, it seeks to find the optimal parameters $\theta^*$ of a classifier $f_\theta(x)$ that are robust to perturbations $\delta$ within a norm ball, *i.e.,*

$$\min_\theta \mathbb{E}_{(x,y)\sim\mathcal{D}}\Big[\max_{\|\delta\|\leq\epsilon} \ell(f_\theta(x+\delta), y)\Big], \tag{1}$$

where $\ell$ is a loss function (*e.g.,* cross-entropy loss) and $\mathcal{D} = \{x, y\}_{i=1}^{|\mathcal{D}|}$ is the training data. (Madry et al., 2017) show that this saddle-point problem can be solved reliably with SGD for the outer minimization and with PGD (Madry et al., 2017) for the inner maximization. PGD starts at randomly initialized perturbations in the $\epsilon$-ball and iteratively applies a gradient ascent update followed by a projecting onto the ball $P$ times, *i.e.,* $\forall p \in [1, p]$,

$$\delta^{(p)} = \Pi_{\|\delta\|\leq\epsilon}\big(\delta^{(p-1)} + \alpha\nabla_x\ell(f(x+\delta^{(p-1)}), y)\big), \tag{2}$$

where $\Pi_{\|\delta\|\leq\epsilon}$ is the projection operator into the $P$-ball and $\alpha$ is the learning rate. $P$-step PGD adversarial training is an order of magnitude more expensive than standard robust training as the $P$-step PGD requires $P$ forward-backward passes through the network, while the standard SGD update requires only one. Next, we review work on curbing this overhead for more scalable AT.

**Scalable AT.** Different optimizations have been proposed to mitigate the cost of the additional $P$ forward-backward passes for AEs generation, including (i) replacing multi-step PGD with a single-step FGSM, enabling simultaneous update of model weights and input perturbation through a single backward pass (Shafahi et al., 2019); (ii) omitting redundant computations during PGD-based backpropagation for additional speedup (Zhang et al., 2019); (iii) combining FGSM adversarial training with random initialization to overcome multi-step PGD (Wong et al., 2020); and (iv) early stopping of the training procedure (Rice et al., 2020). These approaches aim at mitigating the complexity of PGD computation but also come with limitations. For instance, the approach in (i) is vulnerable to 'stale gradients' (Dutta et al., 2018), *i.e.,* the samples do not guarantee the optimization of both the min and max programs. To address this, FreeLB (Zhu et al., 2020) proposes accumulating model parameter gradients over multiple batches. However, due to the limited number of fine-tuning epochs, FreeLB also necessitates multiple steps of PGD, resulting in several rounds of backpropagation. In our work, we achieve scalability through an alternative method, *i.e.,* by leveraging the manifold conjecture to reduce the length of backward-forward chains in PGD.

**Manifold-based defenses.** The manifold hypothesis stands as one of the most compelling explanations for the susceptibility of deep neural networks to adversarial samples (Tanay & Griffin, 2016; Gilmer et al., 2018; Shamir et al., 2021). This hypothesis fundamentally posits that data resides on a low-dimensional manifold within a high-dimensional representation space, and that a network, during training, learns to approximate this manifold. Consequently, an off-manifold sample, deviating from this foundational manifold, leads to undefined behavior in the network. The off-manifold has inspired a novel line of defenses against adversarial attacks on images (Samangouei et al., 2018; Meng & Chen, 2017; Song et al., 2017). Mainly, these methods first approximate the data manifold via eigenvector decomposition (Xiao et al., 2022) or by learning a latent-space generative model (*e.g.,* GAN (Samangouei et al., 2018), VAE (Schott et al., 2018)). Then, at test time, they project AEs to the manifold before classification. (Minh & Luu, 2022) report a similar phenomenon in the contextualized embedding space induced by pre-trained LMs and show that analogous manifold-based defense techniques lead to improved generalization and robustness. While these approaches leverage the manifold conjecture for test-time defense, we use it during training to improve scalability and eventually robustness. Our work is inspired by the observation that different layers of a deep neural network exhibit diverse intrinsic dimensions (ID). Specifically, LMs demonstrate a monotonically decreasing characteristic. Our approach leverages this observation by recognizing that AEs at a layer with low ID will also be AEs at previous layers with higher ID. SMAAT builds upon these insights and identifies the layer with the lowest ID to generate AEs. This not only enhances robustness by simplifying the generation of off-manifold AEs but also improves scalability through shorter forward-backward chains for PGD.

## 3 APPROACH

Achieving robustness through AT relies on training with as many off-manifold AEs as possible. Further, AT can be individually applied to each layer of a network to achieve better robustness. This would, however, significantly hinder the scalability of a method. The key idea behind SMAAT is to generate strong AEs with the lowest computational cost. Our main intuition is that some layers of a network may capture most of the AEs. Thus, the AT cost can be minimized by identifying those layers and hardening them. Our hypothesis is that when the intrinsic dimension of feature representation monotonically decreases across layers, the layer with the lowest intrinsic dimension will include the majority of AEs from previous layers. In order to maintain high robustness with scalability, SMAAT chooses to perturb the highest layer with the lowest intrinsic dimension, which leads to shorter gradient-based attack forward-backward chains and accelerates AE generation as illustrated in Fig. 2.

### 3.1 PROBLEM FORMULATION

Consider a deep neural network classifier $f_\theta$ with $n$ layers and parameters $\theta = \{\theta^{(i)}\}_{i=1}^n$. Given a dataset $\mathcal{D} = \{x_i, y_i\}_{i=1}^{|\mathcal{D}|}$, our goal is to *efficiently* search for the parameters $\theta$ of a classifier $f_\theta$ that result in the highest *robustness* under the worst-case perturbation $\delta^* \in \{\|\delta\| < \epsilon\}$ applied to the input. Building upon the manifold conjecture, which posits that AT with off-manifold examples enhances robustness, whereas AT with on-manifold examples improves generalization, as discussed in Stutz et al.'s work (Stutz et al., 2019), we can augment the AT objective as follows:

$$\min_\theta \quad \mathbb{E}_{(x,y)\sim\mathcal{D}}\Big[ \max_{\|\delta\|\leq\epsilon} \ell(f_\theta(x+\delta), y)\Big]$$
$$\text{s.t.} \quad (x+\delta) \text{ is off-manifold}. \tag{3}$$

Nonetheless, this formulation is conservative as it overlooks the input space of the intermediate layers. In other words, AEs that are off-manifold in any layer contribute to improved robustness, we propose a relaxation to the ojective above:

$$\min_\theta \quad \mathbb{E}_{(x,y)\sim\mathcal{D}}\Big[ \max_{\|\delta\|\leq\epsilon} \ell(f_\theta(x+\delta), y)\Big]$$
$$\text{s.t.} \quad f_{\theta[0,l]}(x+\delta) \text{ is off-manifold in } l^{th} \text{ layer}, \exists l \in [0, n]. \tag{4}$$

where $f_{\theta[i,j]}$ denotes the nested transformation spanning layers $i$ to $j$, *i.e.*, $f_{\theta[i,j]} = f_{\theta(i)} \circ \cdots \circ f_{\theta(j)}$ and $f_\theta$ denotes $f_{\theta[0,L]}$. An AE can be deemed off-manifold in $l^{th}$ layer if its projection error onto the corresponding manifold of the representation space is large. Assuming that the manifold generated between layers is linear and can be approximated using Singular Value Decomposition (SVD), the projection error for AEs can be straightforwdly computed. The SVD basis vectors obtained in the input space of $l^{th}$ layer form an orthonormal set as they are obtained by normalizing the top eigenvectors of the symmetric matrix $f_{\theta[0,l-1]}(X)f_{\theta[0,l-1]}(X)^T$. With $U_l$ representing this basis, the projection error of the sample $f_{\theta[0,l-1]}(x)$ can be computed as $\left(f_{\theta[0,l-1]}(x) - U_l^{k_l}U_l^{k_l T}f_{\theta[0,l-1]}(x)\right)$ where $U_l^{k_l}$ represents the corresponding eigenvectors of the top-$k_l$ eigenvalues required for projecting training samples with limited error in $l^{th}$ layer. Thus, we can assert that, $\forall x \in \mathcal{D}$:

$$f_{\theta[0,l-1]}(x+\delta) \text{ is off-manifold} \Leftrightarrow \|(I-U_l^{k_l}U_l^{k_l T})(f_{\theta[0,l-1]}(x+\delta))\| > 0 \Leftrightarrow \|(I-U_l^{k_l}U_l^{k_l T})\delta_l\| > 0, \tag{5}$$

where $\delta_l$ is the perturbation to the $l^{th}$ layer within the $\epsilon_l$-norm ball, with $\epsilon_l \leq |\lambda_{\max}\{J_{f_{\theta[0,l-1]}}(x)\}|\epsilon$ (see Appendix B). The last equivalences holds as, by default, $f_{\theta[0,l-1]}(x)$ is on-manifold, *i.e.*, $(U_l^{k_l}U_l^{k_l T})f_{\theta[0,l-1]}(x) \approx 0$.

### 3.2 SCALABLE MANIFOLD AWARE ADVERSARIAL TRAINING

To achieve the best possible robustness, our objective is to identify the optimal layer, denoted as $l^*$, which encompasses all off-manifold AEs from previous layers. Simultaneously, we aim to select $l^*$

to be closer to the output to enhance scalability. Formally, the optimum layer satisfies:

$$l^* = \max_i \left\{ i \in [1,n] \middle| \forall x \in \mathcal{D} : f_{\theta[0,i-1]}(x+\delta) \text{ is } \textit{off-manifold} \text{ in the } (i-1)^{\text{th}}\text{layer} \Rightarrow \right.$$
$$\left. f_{\theta[0,i]}(x+\delta) \text{ is } \textit{off-manifold} \text{ in the } i^{\text{th}} \text{ layer} \right\}. \tag{6}$$

In case of a linear manifold (Eq. 5), the combinatorial program above becomes:

$$l^* = \max_i \left\{ i \in [1,n] \middle| \forall x \in \mathcal{D} : \left| \|(I - U_{i-1}^{k_{i-1}} U_{i-1}^{k_{i-1}T})\delta_{i-1}\| > 0 \Rightarrow \|(I - U_i^{k_i} U_i^{k_i T})\delta_i\| > 0 \right\} \right. . \tag{7}$$

To satisfy Eq. 7, a sufficient condition is as follows:

$$l^* = \max_i \left\{ i \in [1,n] \middle| \forall x \in \mathcal{D} : \left| \|(I - U_{i-1}^{k_{i-1}} U_{i-1}^{k_{i-1}T})\delta_{i-1}\| < \|(I - U_i^{k_i} U_i^{k_i T})\delta_i\| \right\} \right. . \tag{8}$$

**Theorem 3.1** *If $f$ is differentiable with Lipschitz continuous gradients, and the intrinsic dimension across layers $i \in [1,n]$ satisfies $k_{i-1} < k_i$, it follows that*

$$\|(I - U_{i-1}^{k_{i-1}} U_{i-1}^{k_{i-1}T})\delta_{i-1}\| < \|(I - U_i^{k_i} U_i^{k_i T})\delta_i\|. \tag{9}$$

*Proof Sketch:* When we constrain the $\delta_i$ with the Jacobian of the previous layer input (as detailed in Appendix B), and under the assumption that $f$ is differentiable with Lipschitz continuous gradients, the projection error becomes dependent on the quantity $\|(I - U_{i-1}^{k_{i-1}} U_{i-1}^{k_{i-1}T})\|$ which is inversely proportional to the dimension of matrix $U_{i-1}^{k_{i-1}}$ (as explained in Appendix C).

Hence, following Theorem 3.1, since the ID of the layer $l$ can be defined by $k_l$, for a given layer $l$ of the network, if the ID of the input space exceeds the ID of the output space, off-manifold AEs in the input space of $l$ will remain off-manifold at the output of $l$. In light of this, the objective in Eq. 4 can be relaxed as follows:

$$\min_\theta \quad \mathbb{E}_{(x,y)\sim\mathcal{D}} \left[ \max_{\|\delta\|\leq\epsilon} \ell(f_\theta(x+\delta), y) \right]$$
$$\text{s.t.} \quad f_{\theta[0,l]}(x+\delta) \textit{ is off-manifold in } l^{\text{th}} \textit{ layer}, \exists l \in [l^*, n]. \tag{10}$$
$$\text{s.t.} \quad ID(i-1) > ID(i); \quad \forall i < l^*,$$

where $ID(i)$ is the intrinsic dimension of $i^{\text{th}}$ layer.

Essentially, this equation reduces the dependency on the layers before $l^*$ by assessing whether the samples are off-manifold or not in layer $l^*$ and the subsequent layers. Under the assumption that, for each AE, $f(x+\delta)$ has a close sample in layer $l^*$ such that $f_{\theta0,l^*-1}(x+\delta) \approx f_{\theta0,l^*-1}(x) + \delta_{l^*}$, AT can effectively operate in layer $l^*$. Based on this, the objective of SMAAT can be expressed as follows:

$$\min_\theta \quad \mathbb{E}_{(x,y)\sim\mathcal{D}} \left[ \max_{l^* \in [1,n], \|\delta_{l^*}\|\leq\epsilon_l^*} \ell\left( f_{\theta[l^*,n]}\left( f_{\theta[0,l^*-1]}(x) + \delta_l^* \right), y \right) \right],$$
$$\text{s.t.} \quad ID(i-1) > ID(i); \quad \forall i < l^*, \tag{11}$$

In essence, SMAAT applies AT in the upper layers to ensure scalability while preserving off-manifold AEs from the lower layers. The retention of AEs in the upper layer relies on the prior layers displaying a monotonically decreasing intrinsic dimension behavior. With this condition met, SMAAT effectively reduces the length of the forward-backward chain required for PGD. The following section empirically investigates the validity of this constraint for NLP models.

### 3.3 EMPIRICAL SEARCH FOR THE OPTIMAL LAYER $l^*$

We empirically tackle the inner maximization program over layer $l$ at the start of training and maintain it consistently for all samples, as all layers with $l < l^*$ are treated as frozen. In order to pinpoint

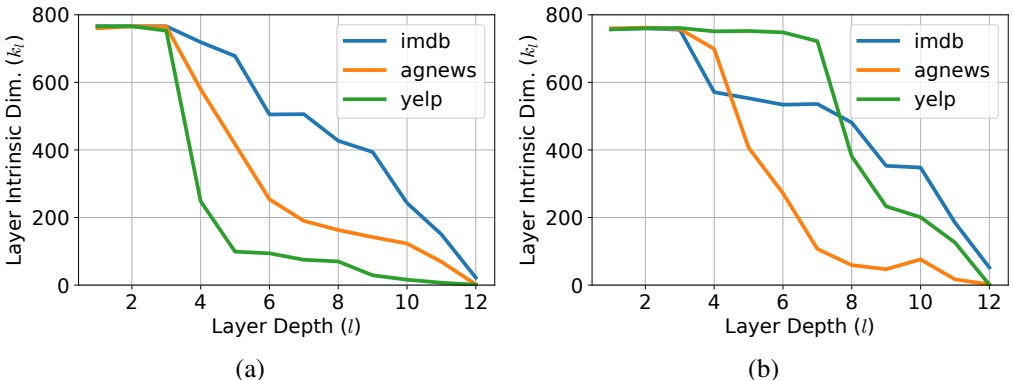

Figure 3: Examination of the intrinsic dimensionality of (a) BERT and (b) RoBERTa layers reveals that the final layers in both models exhibit remarkably low intrinsic dimensionality.

the optimal layer $l^*$ that meets the condition in Eq. 11, we conduct an empirical examination of the ID across the layers of the network. It has been observed that transformer-based NLP models often exhibit a monotonically decreasing trend in ID across their layers (Ethayarajh, 2019). To validate this behavior, we have generated plots of the ID for BERT (Devlin et al., 2018) and RoBERTa (Liu et al., 2019) models across their respective layers. Although various techniques can be employed to measure the ID of a network (Facco et al., 2017; Ceruti et al., 2012; Johnsson et al., 2014; Campadelli et al., 2015), in accordance with our earlier assumptions on the linearity of data manifold, we utilized SVD. The ID of a layer is determined as the point at which the average projection error becomes less than a specified threshold as defined below:

$$\text{ID}(l) = \min_{k}\{\frac{\sum_{x \in \mathcal{D}} 1 - cos(f_{\theta^0, l}(x), U_l^{k_l} U_l^{k_l T} f_{\theta^0, l}(x))}{\|D\|} < thr\}. \tag{12}$$

Here, the function $\cos()$ represents the cosine similarity between the original and projected samples, and $thr$ denotes the specified threshold value. For the duration of this paper, we use $thr = 0.1$.

In Fig. 3, we empirically obtain the ID of BERT and RoBERTa models across different layers using IMDB, AGNEWS, and YELP datasets. Consistent with the literature, the conditions in Theorem 3.1 are met when setting $l^* = n$. This signifies that SMAAT can effectively operate the last layer of these models. Analysis on vision and decoder models can be seen in Appendix Sections D and E.

### 3.4 COMPLEXITY

SMAAT proceeds in two steps. In the first step, it finds the optimal layer $l^*$ to perturb which is done once per model and per task. and incur marginal overhead. Next, SMAAT adversarially trains $l^{*\text{th}}$ layer of the model by generating AEs from layer $l^*$ instead of from the input layer. When $K$-step PGD attack is used, this results in $K$ forward-backward passes with length $(n - l^* + 1)$ instead of $n$. The run-time of every forward/backward pass depends on the layer dimensionality, *i.e.,* $O(d_{l-1} \times d_l)$ for the $l^{\text{th}}$ layer. Overall, the complexity of one SMAAT forward-backward is $\mathcal{O}((n - l^* + 1) \max_{l \in [l^*, n]}(d_l)^2))$. As a result, SMAAT is more efficient than classical AT by a factor of $\mathcal{O}\Big(k \times l^* \times (\max_{i \in [1,n]}(d_i)^2 - \max_{j \in [l^*, n]}(d_j)^2)\Big)$. In the case of BERT and ROBERT, $l^*$ is equal to $= n$. The total run-time can be simplified to $\mathcal{O}(k \times (d_n)^2)$ where $d_n$ is the number of classes. Typical NLP tasks consist of less than five classes (Wang et al., 2018a) which makes the factor of classes small enough to be negligible. Hence, SMAAT enhances the efficiency of the AEs generation process by a factor of $l \times O(\max(d_l))$. This improvement practically eliminates the cost of the AE generation process.

### 4 EVALUATION

We evaluate SMAAT using three tasks across two models. We employ the established evaluation setting commonly used by previous baseline methods.

Table 1: Robustness and generalization results on AGNEWS, IMDB, and YELP datasets. We report the clean accuracy (CA) and the robust accuracy under PWWS (PW), TextFooler (TF), and Bert-Attack (BA) attacks, along with the average robust accuracy (AR) across the three attacks. To facilitate comparison, we have added a *Best Score* row to each model, indicating the highest score from the baselines for each column. The best performance for each model is highlighted in **bold**.

| Model | Defense | AGNEWS | | | | | IMDB | | | | | YELP | | | | |
|---|---|---|---|---|---|---|---|---|---|---|---|---|---|---|---|---|
| | | CA | PW | TF | BA | AR | CA | PW | TF | BA | AR | CA | PW | TF | BA | AR |
| BERT | Standard | 94.5 | 36.9 | 28.1 | 37.5 | 34.2 | 92.2 | 15.0 | 5.8 | 5.4 | 8.7 | **97.0** | 12.2 | 6.5 | 5.3 | 8.0 |
| | ASCC | 91.6 | 32.8 | 31.4 | 32.1 | 32.1 | 88.5 | 15.1 | 12.4 | 11.2 | 12.9 | 91.5 | 19.4 | 15.7 | 12.2 | 15.8 |
| | DNE | 94.1 | 34.0 | 33.6 | 52.3 | 40.0 | 90.0 | 25.7 | 23.0 | 20.6 | 23.1 | 94.0 | 33.3 | 31.2 | 43.8 | 36.1 |
| | FreeLB++ | **95.1** | 47.9 | 51.5 | 41.8 | 47.1 | **93.2** | 12.5 | 45.3 | 39.9 | 32.6 | 95.6 | 19.3 | 8.8 | 3.7 | 10.6 |
| | SAFER | 94.4 | 39.3 | 35.5 | 42.3 | 39.0 | 92.3 | 41.4 | 39.1 | 30.7 | 37.1 | 95.4 | 29.8 | 25.8 | 23.7 | 26.4 |
| | TMD | 94.3 | 70.0 | 50.0 | 55.2 | 58.4 | 92.2 | 38.7 | 44.2 | 33.7 | 38.9 | 95.2 | 36.8 | 40.9 | 28.6 | 35.4 |
| | RSMI | 92.7 | **76.1** | 63.2 | NA[1] | NA | 92.2 | 58.7 | 56.4 | NA[1] | NA | 95.4 | 45.3 | 52.3 | NA[1] | NA |
| | *Best Score* | *95.1* | *76.1* | *63.2* | *55.2* | *64.8* | *93.2* | *58.7* | *56.4* | *39.9* | *51.7* | *97.0* | *45.3* | *52.3* | *43.8* | *47.1* |
| | **SMAAT (Ours)** | 94.6 | 73.5 | **72.2** | **74.7** | 73.5 | 92.2 | **63.6** | **77.9** | **60.8** | 67.4 | **97.0** | **77.1** | **77.9** | **72.8** | 75.9 |
| RoBERTa | Standard | 94.7 | 30.6 | 23.9 | 37.1 | 30.5 | 94.0 | 8.7 | 2.1 | 0.6 | 3.8 | 97.9 | 23.1 | 14.9 | 9.0 | 15.7 |
| | ASCC | 92.6 | 48.1 | 41.0 | 49.1 | 46.1 | 92.6 | 23.1 | 13.5 | 11.8 | 16.1 | 95.4 | 15.0 | 8.6 | 4.5 | 9.4 |
| | DNE | 94.9 | 58.0 | 46.5 | 54.5 | 53.0 | 94.2 | 48.8 | 26.9 | 16.0 | 30.6 | 96.8 | 64.4 | 64.0 | 45.2 | 57.9 |
| | FreeLB++ | **95.6** | 61.0 | 49.8 | 56.6 | 55.8 | **94.3** | 33.6 | 14.6 | 6.1 | 18.1 | 97.0 | 38.6 | 46.0 | 35.2 | 39.9 |
| | SAFER | 94.6 | 68.9 | 49.3 | 46.1 | 54.8 | 93.9 | 58.4 | 47.1 | 40.6 | 46.8 | 96.6 | 65.6 | 67.9 | 48.3 | 60.6 |
| | TMD | 95.0 | 68.3 | 54.0 | 56.7 | 59.7 | 93.3 | 60.5 | 66.8 | 51.6 | 59.6 | 96.6 | 68.9 | 70.9 | 51.0 | 63.6 |
| | RSMI | 94.3 | **81.9** | 74.1 | NA[1] | NA | 93.0 | 76.2 | 73.4 | NA[1] | NA | 96.3 | 68.9 | 65.9 | NA[1] | NA |
| | *Best Score* | *95.6* | *81.9* | *74.1* | *56.7* | *70.9* | *94.3* | *76.2* | *73.4* | *51.6* | *67.1* | *97.0* | *68.9* | *70.9* | *51.0* | *63.6* |
| | **SMAAT (Ours)** | 94.6 | 75.6 | **75.1** | **79.9** | 76.9 | 93.5 | **77.1** | **78.5** | **63.2** | 72.9 | **98.0** | **85.4** | **86.4** | **76.0** | 82.6 |

## 4.1 EXPERIMENTAL SETTINGS

**Datasets.** We use three classification benchmarks: (1) AGNEWS (Zhang et al., 2015b), (2) IMDB (Maas et al., 2011), and (3) YELP (Zhang et al., 2015a). AGNEWS contains over 120k samples across four categories: World, Sports, Business, and Sci/Tech. IMDB consists of 50k movie reviews with binary sentiment labels. Due to resource limitations, we use a subset of 63k samples from YELP binary sentiment classification dataset.

**Base model.** We use BERT-base-cased (Devlin et al., 2019) and RoBERTa-base-cased (Liu et al., 2019) which are 12 layered models.

**Adversarial Attacks.** We assess the robustness of our approach against three word substitution based input space attacks: (1) PWWS (Ren et al., 2019a) (synonym based), (2) TextFooler (Jin et al., 2020) (neighbor based), and (3) BERT-Attack (Li et al., 2020a) (masked language model based). All attacks are conducted using the TextAttack framework (Morris et al., 2020) and following the settings introduced by Li et al. (2021a).

**Baselines.** We compare SMAAT to several baselines including standard (non-adversarial) training, and six baselines from three families of defenses: (1) Input space AT (ASCC, (Dong et al., 2021), DNE (Zhou et al., 2021)), (2) Embedding space AT (FreeLB++ (Li et al., 2021a)), and (3) Certified defenses (SAFER (Ye et al., 2020), TMD (Minh & Luu, 2022), RSMI (Minh & Luu, 2022)). FreeLB++ focuses on scalability and robustness by minimizing the number of PGD steps and applying AT in the initial layer. TMD leverages manifold features by projecting samples back to the manifold in the last layer.

**Implementation details.** As explained above, we generate AEs in the last layer. Specifically, we perturb the [CLS] embeddings before the classifier layer ($h^l$). We train the last layer of $f_\theta$ for ten epochs with varying attack strength $\epsilon$ from 0.1 to 0.8 and learning rate $\tau = 0.1$. Notably, these $\epsilon$ and $\tau$ values are higher than those typically used in standard AT, as we are only training the network with one layer. For more details about the experimental setup, please refer to Appendix Sec. F.

## 4.2 RESULTS

**Robustness and generalization results** Table 1 compares the results of baselines and SMAAT on robustness and generalization whereas generalization is measured as the performance difference be-

---

[1]As mentioned in the original paper, the current implementation of RSMI takes around 2k times more time than the TextFooler algorithm to generate a single adversarial example with BERT-Attack. Due to this significant time difference, it becomes infeasible to test RSMI with BERT-Attack.

tween an adversarially trained system and the clean accuracy (CA). On average, SMAAT demonstrates superior robustness over both datasets. Specifically, it achieves an improvement of 8.6%, 15.7%, and 28.8% over the best score for the BERT model, and 6.0%, 5.8%, and 19.0% for the RoBERTa model on the AGNEWS, IMDB, and YELP datasets, respectively. This observation supports the insight that shifting to higher layers under the constraint of monotonically decreasing ranks leads to increased robustness. Note that FreeLB++, which perturbs the first layer, showed the best generalization in four out of six cases, as it produces mostly on-manifold examples (the first layer has a high dimension, as illustrated in Fig. 3). SMAAT maintains generalization in five out of six cases and only shows 0.5 drop in performance in the case of RoBERTa with the IMDB dataset.

TMD is another manifold-aware AT method that directly estimates the manifold and projects the input samples onto it prior to classification. This may result in inaccuracies due to manifold-estimation errors. SMAAT consistently performed better than TMD by a large margin. From an attack perspective, RSMI demonstrates superior robustness against PWWS, a synonym-based word substitution attack, in both models. This phenomenon can be attributed to the behavior of the masked inference (MI) component in RSMI, which mimics a synonym-substitution based defense approach without requiring an explicit synonym set, as noted in the original paper (Minh & Luu, 2022).

Table 2: Run-time results on IMDB dataset. Mean and standard deviation are computed over ten runs. We observed similar results across other tasks and models.

|  | Standard | ASCC | DNE | FreeLB++ | SAFER | RSMI | SMAAT |
|---|---|---|---|---|---|---|---|
| Training (min/epoch) | 5.1 ±0.1 | 25.7 ±0.3 | 15.2 ±0.9 | 15.6 ±0.5 | 8.2 ±0.6 | 15.4 ±0.3 | 5.2 ±0.2 |
| Inference (msec/sample) | 2.4 ±0.1 | 41.4 ±0.2 | 4.0 ±0.2 | 2.4 ±0.0 | 2.4 ±0.0 | 5.6 ±0.4 | 2.4 ±0.1 |

**Runtime efficiency.** In Table 2, we provide details on the training time per epoch and inference time per instance for the IMDB dataset using the BERT model. We focus on reporting these values for BERT since both models share the same architecture, with the main difference lying in their pre-training processes. SMAAT have comparable efficiency to standard training and is on average, 3 times faster during training and 4.6 times more efficient during testing. This is due to the fact that SMAAT performs AT in the last layer which results in short backpropagation chains when creating AEs. This stands in contrast to methods like FreeLB++, which inject noise in the first layer, leading to the computation of longer AEs' backpropagation chains at every iteration. Even when we substitute FreeLB++ with FGSM-based scalable AT methods, which have previously been shown to underperform FreeLB (Zhu et al., 2020; Li et al., 2021a), their runtime remains less efficient than that of SMAAT since they require a full-depth backpropagation to create AEs. Additionally, certified defense baselines (SAFER, RSMI) and input space attacks (ASCC, DNE) are also time-consuming as they either require mapping samples into the manifold or performing an extensive search over word substitutes.

**AT on Intermediate Layers.** The key idea behind SMAAT is that if a model exhibits monotonically decreasing ID behavior, AEs generated in higher layers encompass those from the previous layers. Therefore, higher-layers are expected to yield improved robustness. To empirically validate this, we apply AT to layers 0, 2, 4, 6, 8, 10, 11, and 12. During training, for each layer we conduct a separate grid search for hyperparameters, by varying the learning rate from 0.01 to 0.00001 and $\epsilon$ from 0.1 to 0.001. We keep the number of PGD steps to three 3 and the number of training epochs to 5.

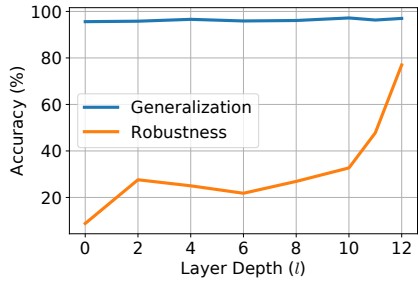

Figure 4: Effect of AT at different layers. Robustness improves as AT is applied to higher layers.

We employ the BERT model on the YELP dataset, assessing both the model's generalization on clean samples and its robustness using the TextFooler attack. The results depicted in Fig. 4 unambiguously demonstrate that applying AT in the higher layers yields superior robustness compared to the initial layers. Moreover, our analysis of

Table 3: The average robustness results of the standard, FreeLB++, and SMAAT models when subjected to the targeted-PGD attack and the targeted "feature-level" attack on the AGNEWS, IMDB, and YELP datasets. The results clearly demonstrate that SMAAT enhances model robustness in the face of targeted PGD based attacks.

| Model | BERT | | | RoBERTa | | |
|---|---|---|---|---|---|---|
| | AGNEWS | IMDB | YELP | AGNEWS | IMDB | YELP |
| Standard | 8.2 | 0.1 | 6.9 | 40.0 | 19.3 | 79.2 |
| FreeLB++ | 20.1 | 10.6 | 20.2 | 62.4 | 46.7 | 76.4 |
| SMAAT (Ours) | 49.9 | 18.0 | 28.4 | 77.1 | 59.4 | 83.5 |

the manifold behavior of AEs reveals that AEs deviate further from the manifold as we progress to higher layers (refer to Appendix H for detailed insights).

**Effect of the attack.** In Table 1, the attack techniques evaluated primarily employ word swapping to create AEs. To further assess the effectiveness of SMAAT against more potent attacks, we apply the PGD attack directly to the token embeddings. The PGD attack is more powerful because it doesn't require mapping embeddings back to the input space, as other attacks do (Guo et al., 2021). We use a 50-step targeted PGD attack with the $\epsilon$ value set to 0.003. Additionally, following (Tramer et al., 2020), we applied a more aggressive attack by combining PGD with the "feature-level" attack (Sabour et al., 2015). This attack targets intermediate layer representations of other classes, posing a potent challenge to SMAAT, which focuses on robustifying the last layer. For comparison, we select FreeLB++, another technique designed for scalable AT. However, it differs from SMAAT in its approach, as it operates in the initial layers rather than the higher ones. The results in Table 3 show that the efficacy of SMAAT is not limited to word swapping based attacks, and it is also effective against the more powerful PGD attack. Supplementary results on GLUE and advGLUE benchmarks provide further evidence of SMAAT's effectiveness. Refer to Appendix G for detailed information.

## 5 CONCLUSION

In this paper, we introduce a manifold-aware approach designed to enhance the scalability, and robustness of adversarially trained deep neural networks while maintaining generalization. Our approach involves generating adversarial examples in higher layers, guided by their monotonically decreasing rank behavior. This strategy enables faster synthesis with shorter backpropagation chains and enhances robustness by capturing off-manifold data. As a result, we achieve state-of-the-art performance on several tasks, excelling in both runtime efficiency and robustness, all while maintaining comparable levels of generalization.

## LIMITATIONS

**Incompatibility with Vision Models**: SMAAT cannot be equally effective on vision models because they lack monotonically decreasing ID behavior. Vision models typically exhibit a "hunchback shape" behavior in terms of intrinsic dimension, characterized by an initial increase followed by a gradual decrease in ID towards the final layers (Ansuini et al., 2019). A more detailed analysis of vision models can be found in Appendix D.

**Encoder Model Focus**: All experiments conducted in this study assume encoder models and primarily focus on classification tasks. Extending SMAAT to other architectures such as decoder-only and encoder-decoder models could provide valuable insights into its adaptability across various model architectures. Our preliminary assessment on the decode-only LLAMA-2 model indicates a similarity in its ID behavior to vision models. Additional results can be found in Appendix E.

**Lack of Out-of-Distribution Evaluation**: While the primary focus of this work is on robustness and generalization concerning AEs and clean samples, evaluating the approach's performance on out-of-distribution data could offer a different perspective on its effectiveness.

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

## A   SUPPLEMENTARY MATERIAL

We propose SMAAT an adversarial training algorithm that doesn't only optimize for better robustness/generalization as performed conventionally, but also for scalability.

## B   PROOF: PERTURBATION LIMIT

Under the assumption that $f$ is differentiable, we prove $\delta_l$ is limited by $\epsilon_l$:

$$f_{\theta[1,l-1]}(x + \delta) \approx f_{\theta[1,l-1]}(x) + \delta_l$$
$$\text{s.t.} \|\delta_l\| \leq \epsilon_l$$

Since $f$ is continuous and differentiable, we use Taylor expansion to approximate $f_{\theta[1,l-1]}(x + \delta)$ on the interval $[x, x + \delta]$: $f_{\theta[1,l-1]}(x + \delta) = f_{\theta[1,l-1]}(x) + J_{f_{\theta[1,l-1]}}(x)\delta$, where $J_{f_{\theta[1,l-1]}}(x) = \frac{\partial f_{\theta[1,l-1]}(x)}{\partial x}$.

We denote by $\delta_l$ perturbation in the feature space of $f_{\theta[1,l-1]}$ and $\delta_l = J_{f_{\theta[1,l-1]}}\delta$. Since the $\|\delta\| < \epsilon$:
$\|\delta_l\| < \|\lambda_{\max}\{J_{f_{\theta[1,l-1]}}(x)\}\|\epsilon = \epsilon_l$,
where $\lambda_{\max}$ is the maximum eigenvalue.

## C   PROOF: THEOREM 3.1

If $f$ is differentiable with Lipschitz continuous gradients.

$$k_{i-1} > k_i \rightarrow \{\|(I - U_{i-1}^{k_{i-1}} U_{i-1}^{k_{i-1}}{}^T)\delta_{i-1}\| < \|(I - U_i^{k_i} U_i^{k_i}{}^T)\delta_i\|\} \tag{13}$$

For ease of exposition, we prove the contrapositive of Equation 13. That is:

$$\{\|(I - U_{i-1}^{k_{i-1}} U_{i-1}^{k_{i-1}}{}^T)\delta_{i-1}\| > \|(I - U_i^{k_i} U_i^{k_i}{}^T)\delta_i\|\} \rightarrow k_{i-1} < k_i \tag{14}$$

First, we substitute $\delta_i$ with the upper bound $|\lambda_{\max}\{J_{f_{\theta[1,i-1]}}(x)\}|\epsilon$ value as

$$\{\|(I - U_{i-1}^{k_{i-1}} U_{i-1}^{k_{i-1}}{}^T)\|\lambda_{\max}\{J_{f_{\theta[1,i-2]}}(x)\}|\epsilon > \|(I - U_i^{k_i} U_i^{k_i}{}^T)\|\lambda_{\max}\{J_{f_{\theta[1,i-1]}}(x)\}|\epsilon\}$$
$$\Rightarrow \frac{\|(I - U_{i-1}^{k_{i-1}} U_{i-1}^{k_{i-1}}{}^T)\|}{\|(I - U_i^{k_i} U_i^{k_i}{}^T)\|} > \frac{\|\lambda_{\max}\{\|J_{f_{\theta[1,i-1]}}(x)\|\}\epsilon\|}{\|\lambda_{\max}\{\|J_{f_{\theta[1,i-2]}}(x)\|\}\epsilon\|} \tag{15}$$

Under the assumption $f$ has Lipschitz continuous gradients, since the numerator of the last term in Eq. (15) involves one more layer than the denominator, $\frac{\|\lambda_{\max}\{\|J_{f_{\theta[1,i-1]}}(x)\|\}\epsilon\|}{\|\lambda_{\max}\{\|J_{f_{\theta[1,i-2]}}(x)\|\}\epsilon\|} > 1$. Hence,

$$\|(I - U_{i-1}^{k_{i-1}} U_{i-1}^{k_{i-1}}{}^T)\| > \|(I - U_i^{k_i} U_i^{k_i}{}^T)\|. \tag{16}$$

Since $I$ is the full rank matrix:

$$\Rightarrow \|U_{i-1}^{k_{i-1}} U_{i-1}^{k_{i-1}}{}^T\| < \|U_i^{k_i} U_i^{k_i}{}^T\|,$$
$$\Rightarrow k_{i-1} < k_i$$

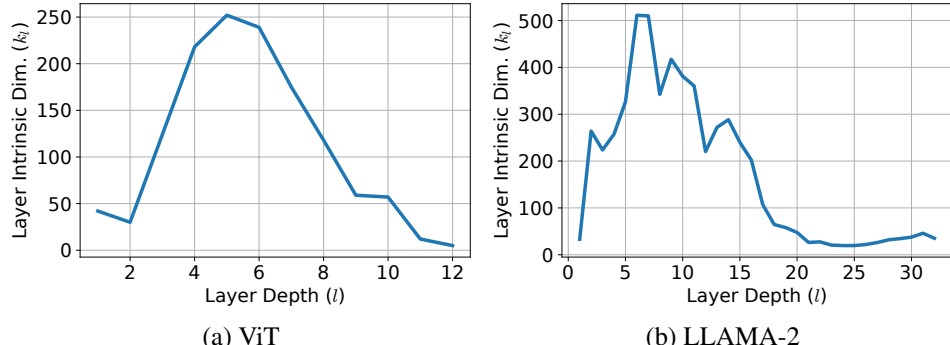

(a) ViT               (b) LLAMA-2

Figure 5: Examination of the ID of (a) the ViT model trained on the CIFAR-10 dataset and (b) the LLAMA-2-7b model trained on the SST-2 dataset. The results indicate that for these models, AT needs to be applied in the initial layers.

Table 4: Robustness and generalization results of the LLAMA-2 model on the SST-2 dataset with standard, FreeLB++, and SMAAT training. The results indicate that SMAAT does not provide a robustness advantage over the standard approach on the LLAMA-2 model.

| Dataset | Standard | FreeLB++ | SMAAT (Ours) |
|---|---|---|---|
| SST-2 | 95.5 | 95.1 | 95.1 |
| Adversarial SST-2 | 69.3 | 69.9 | 69.2 |

## D  APPLICATION OF SMAAT ON VISION MODELS

To verify the "hunchback shape" behavior observed in (Ansuini et al., 2019) for vision models, characterized by an initial increase followed by a gradual decrease in ID towards the final layers, we train a Vision Transformer (ViT) (Dosovitskiy et al., 2020) on the CIFAR-10 dataset. The result depicted in Fig. 5(a) confirms this behavior. The observed pattern suggests that $l^*$ corresponds to the initial layers for vision models. This finding is in line with YOPO (Zhang et al., 2019), which emphasizes the critical role of the initial layers for AT in vision models, and our approach provides an explanation for the significance of these layers. Nevertheless, to ensure validity of our hypothesis we apply AT exclusively to the last layer of the ViT model on the CIFAR-10 dataset. As expected, our results do not demonstrate any improvements in the model's robustness.

## E  APPLICATION OF SMAAT ON LLAMA2

After gaining attention for their effectiveness in text generation, decoder models have become a focus in the literature. In this context, we assess the application of SMAAT on LLAMA-2-7b (Touvron et al., 2023). For this, we train an 8-bit quantized LLAMA model with QLoRA (Dettmers et al., 2023) adaptors on the SST-2 dataset using the SFTTrainer library from HuggingFace (Sup). Given that the initial step of SMAAT involves analyzing the ID of the layers to determine $l^*$ in Eq. 11, we obtain this behavior using the last token representation of each propmt. The measured ID of network layers are presented in Fig. 5(b). This result suggest that the optimal layer for applying AT is the first layer, $l^* = 1$, similar to the ViT model. For the compression, we could only extend our experiments for the FreeLB++ method in addition to SMAAT since implementing all the attacks in Table 1 on the LLAMA-2 model requires changing the design of most of those techniques. More specifically, the certified robustness-based approaches, such as RSMI and SAFER, rely on random masking of input tokens which does not directly apply to causal language models like LLAMA-2. Additionally, for the data augmentation-based approaches like ASCC and DNE, the computational cost would be prohibitively high. The results are presented in Table 4. Our measurements revealed that both FreeLB++ and SMAAT yielded 95% and 69% accuracy on SST-2 and its adversarial version, which was the result obtained on the standard (vanilla) method. This shows that under this setting neither method offers a robustness advantage over the standard.

## F    EXPERIMENTS

In the following, we provide more details about the experiments.

**Datasets.** We evaluate SMAAT on three datasets: AG-News Corpus (AGNEWS) (Zhang et al., 2015b), Internet Movie Database (IMDB) (Maas et al., 2011), and Yelp Review Polarity (YELP) (Zhang et al., 2015a). The AGNEWS dataset contains over 120000 samples, each belonging to one of the four labels: World, Sports, Business, Sci/Tech. The IMDB dataset contains 50000 data samples of movie reviews with binary labels for negative and positive sentiments. The YELP dataset contains nearly 600000 samples of highly polar Yelp reviews with binary labels. However, due to limitations in computing resources, we only use a subset of 63000 samples of the YELP dataset. In addition, we randomly sample 10% of the training set for validation in all datasets. For testing, we use a subset of 1000 test samples from each dataset, following previous work practices. The AGNEWS dataset contains over 120k samples, categorized into four classes: World, Sports, Business, and Sci/Tech. The IMDB dataset consists of 50k movie reviews, each labeled with binary sentiments (positive or negative).

**Base model.** We employed the BERT$_{\text{base-cased}}$ (Devlin et al., 2019) and RoBERTa$_{\text{base-cased}}$ (Liu et al., 2019) models in our experiments. To conduct the evaluations, we utilized the fine-tuned models provided by *TextAttack* from HuggingFace for most datasets, except for the RoBERTa base model fine-tuned for YELP dataset. For the YELP dataset, we created a fine-tuned RoBERTa model by training it for 2 epochs with a learning rate of $\tau = 1e - 05$ and a batch size of 32.

**Adversarial Attacks.** which include the following constraints: (1) The maximum percentage of modified words is set to 0.3 for AGNEWS, 0.1 for IMDB and YELP datasets, respectively. (2) For word replacement, a maximum of 50 candidates are considered for each word. (3) The semantic similarity, measured using the Universal Sentence Encoder (Cer et al., 2018), between the original input and the generated adversarial example must exceed 0.84. PWWS uses word synonyms, TextFooler applies nearest neighbor search in counter-fitting embeddings (Mrkšić et al., 2016), and BERT-Attack utilizes BERT masked language model to generate candidate words.

**Baselines.** For input space adversarial training, we consider Adversarial Sparse Convex Combination (ASCC) (Dong et al., 2021) and Dirichlet Neighborhood Ensemble (DNE) (Zhou et al., 2021). These methods model the perturbation space as the convex hull of word synonyms.

ASCC incorporates an entropy-based sparsity regularizer to capture word substitution geometry more effectively, while DNE expands the convex hull to encompass all synonyms of each word's synonyms, combining Dirichlet sampling and adversarial training to enhance model robustness. In our investigation of embedding space adversarial training, recognized as the most impactful technique for enhancing generalization (Li et al., 2021b), we conduct a thorough analysis of FreeLB++ (Li et al., 2021a) which employs gradient-guided perturbations centered around the most susceptible data points.

For certified defenses, we evaluate SAFER (Ye et al., 2020), TMD (Minh & Luu, 2022), and RSMI (Minh & Luu, 2022). SAFER constructs a set of randomized inputs by performing random synonym substitutions and using the statistical properties of predicted labels to certify robustness. TMD employs infoGAN () to project adversarial examples to the data manifold in the last layer to address the manifold issue. RMSI combines these ideas by applying importance-based masking to tokens and leveraging randomized smoothing in each layer.

**Implementation details.** To train the last layer of $f_\theta$ with adversarial samples, we create adversarial samples using 5-step PGD attacks. During training, we use epsilon values of 0.1, 0.1, and 0.8 for the YELP, AGNEWS, and IMDB datasets, respectively, for the BERT models. For the RoBERTa models, we employ epsilon values of 0.1, 0.6, and 0.03. All models are trained 10 epochs with a learning rate of $\tau = 0.1$.

## G    RESULTS ON GLUE AND ADVGLUE BENCHMARK

The GLUE benchmark (Wang et al., 2018b) is a comprehensive evaluation suite featuring seven diverse NLP tasks to assess model performance. The AdvGLUE benchmark (Wang et al., 2021) is an extension of GLUE, incorporating 17 distinct textual adversarial attacks, covering word-level transformations, sentence-level manipulations, and human-written AEs. This extension ensures a thorough evaluation encompassing various adversarial linguistic phenomena. For our assessment, we

Table 5: Average accuracy of the standard, FreeLB++ and SMAAT models on GLUE and AdvGLUE datasets. The results clearly demonstrate that SMAAT enhances model generalization (GLUE results) and robustness (AdvGLUE results).

| Dataset | BERT | | | RoBERTa | | |
|---|---|---|---|---|---|---|
| | Standard | FreeLB++ | SMAAT (Ours) | Standard | FreeLB++ | SMAAT (Ours) |
| GLUE | 85.9 | 86.3 | 86.3 | 89.3 | 89.6 | 89.7 |
| AdvGLUE | 39.5 | 42.5 | 45.1 | 27.5 | 37.1 | 39.6 |

employ the evaluation sets of four datasets across three different tasks: Sentiment Analysis (SST-2), Duplicate Question Detection (QQP), and Natural Language Inference (QNLI, RTE).

For our evaluation, we compare SMAAT against standard[2] BERT and RoBERTa models and their FreeLB++ incorporated versions. In the case of SMAAT, we conducted a grid search for the learning rate, ranging from 0.1 to 0.001, and the $\epsilon$ value, ranging from 0.8 to 0.01, with 3-PGD steps. As indicated in Table 5, SMAAT exhibits a robustness enhancement of 5.6% and 2.6% for BERT, and 12.1% and 2.5% for RoBERTa, in comparison to the standard and FreeLB++ models, respectively, while maintaining comparable generalization.

## H MANIFOLD BEHAVIOR OF AES

To empirically validate Theorem 3.1, which posits that if the layers exhibit monotonically decreasing ranks, the projection error increases as we move to higher layers. We measure the projection error of both clean samples and AEs across the layers. For AEs, we employ examples generated using TextFooler and PGD attacks from the test set. We leverage the cosine distance metric between the samples and their corresponding representations in the manifold to measure the projection error. Subsequently, we normalize the average projection error of AEs in training set, test set, TextFooler set, and PGD set while using the error of the training set as a reference.

The results in Fig. 6 show that the projection error increases for AEs, which is in line with Theorem 3.1. While there are small drops for some layers, these drops can be related to errors in the manifold estimation.

---

[2]We use the fine-tuned model from *https://huggingface.co/JeremiahZ*

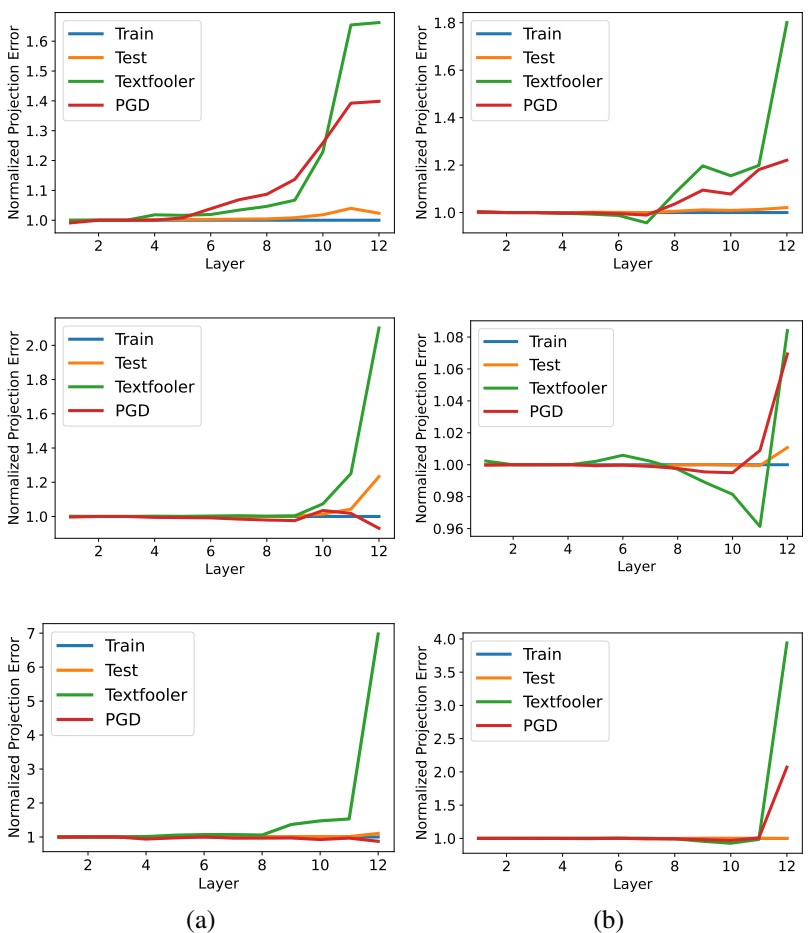

Figure 6: Normalized average projection errors on the AGNEWS, IMDB, and YELP datasets for (a) BERT model and (b) RoBERTa model. As suggested by Theorem 1, the projection error consistently rises for AEs as we progress to higher layers, owing to the layers' monotonically increasing rank.

