# OpenReview forum: "SMAAT: Scalable Manifold-Aware Adversarial Training for Large Language Models"
_ICLR.cc/2024/Conference — Submitted to ICLR 2024_

### Official Review · Reviewer_nXMe · 2023-11-02

**Soundness:** 2 fair
**Presentation:** 3 good
**Contribution:** 2 fair
**Rating:** 5
**Confidence:** 5

**Summary:**

The paper introduces SMAAT, a scalable manifold-aware adversarial training method for large language models. This paper proposes to generate adversarial examples efficiently via the last layer of the model, based on the hypothesis and observation of monotonically decreasing intrinsic dimensionality of the embedding space. The empirical evaluations on AGNEWS, IMDB, and YELP demonstrate improvements in robustness and scalability over previous state-of-the-art methods.

**Strengths:**

+ originality, this paper proposes to adversarially training the last layer of language model to gain adversarial robustness, with seeking scalability.

+ clarity, this paper is clear to follow and easy to read.

**Weaknesses:**

- lack of evidence for scalability: does the hypothesis of intrinsic dimensionality still hold for larger language models?

- ablation study of adversarial training on different layers: since this paper proposes to only fine-tune the last layer, what will happen if we include more layers for adversarial training? do we gain better robustness since more model parameters are included for adversarial training?

- lack of adaptive attack: I strongly recommend the authors to design the adaptive attack [1] to approximate the lower bound of empirical adversarial robustness under their defense. This paper only evaluates model robustness under a simple 5-step PGD attack, which is not enough.

[1] Florian Tramer, Nicholas Carlini, Wieland Brendel, Aleksander Madry, On Adaptive Attacks to Adversarial Example Defenses

**Questions:**

no question

---

> ### Author Response · Authors · 2023-11-21
>
> We thank the reviewer for their feedback and bringing to our attention the adaptive attack work.
>
> > lack of evidence for scalability: does the hypothesis of intrinsic dimensionality still hold for larger language models?
>
> **Response:** We conducted additional experiments on the fine-tuned version of the quantized LLAMA-2 7B model, employing PEFT. Our results indicate that the intrinsic dimensionality of the model displays a non-monotonic pattern. Consistent with our hypothesis, integrating SMAAT into this model results in choosing the first layer to perform adversarial training, which does not provide additional scalability benefits. We nevertheless performed AT at the last layer and, as expected, this did not yield improved robustness.
>
> > ablation study of adversarial training on different layers: since this paper proposes to only fine-tune the last layer, what will happen if we include more layers for adversarial training? do we gain better robustness since more model parameters are included for adversarial training?
>
> **Response:** We evaluated the robustness and generalization impact of adversarial training in the intermediate layers. Our findings are presented in the newly added Figure 4 of the revised paper. In alignment with our main finding, when the adversarially trained layer is positioned closer to the input layer robustness of the model decreases as higher intrinsic dimensionality makes it more difficult to generate off-manifold examples.
>
> > lack of adaptive attack: I strongly recommend the authors to design the adaptive attack [1] to approximate the lower bound of empirical adversarial robustness under their defense. This paper only evaluates model robustness under a simple 5-step PGD attack, which is not enough.
>
> **Response:** To assess the robustness of our method under more targeted attacks, we considered the following three strategies:
> 	1. enhancing the strength of PGD attacks by increasing the steps to 50 and targeting the most vulnerable class, thereby generating attack samples more potent than those utilized during training;
> 	2. implementing targeted PGD attacks focused on the internal representations of an intermediate layer (“feature-level attack” introduced in [[1]](#1)), rather than targeting the output label, to more effectively exploit our approach which primarily bolsters robustness at the final layer.
> The robustness results presented in Table 3 of the revised manuscript are the averaged results of these two attacks instead of the 5-step PGD attack applied earlier.
>
> ## References
> [1]
> Sara Sabour, Yanshuai Cao, Fartash Faghri, and David J Fleet (2015).
> Adversarial manipulation of deep representations
> arXiv preprint arXiv:1511.05122, 2015

---

> > ### Author Response · Authors · 2023-11-23
> >
> > Dear Reviewer nXMe,
> >
> > Thank you once again for your valuable feedback. We have diligently incorporated your suggestions into the manuscript. We are eager to engage in further discussions and address any additional concerns you may have.
> >
> > Best regards

---

### Official Review · Reviewer_6Kya · 2023-11-03

**Soundness:** 3 good
**Presentation:** 3 good
**Contribution:** 3 good
**Rating:** 6
**Confidence:** 3

**Summary:**

This paper proposes a manifold-aware approach to enhance the scalability and efficiency of adversarial training. Specifically, the model generates adversarial examples from higher layers in the deep neural network thereby shortening the gradient-based propagation and accelerating the generation of adversarial examples. Extensive experiments verified the effectiveness of the method.

**Strengths:**

1. Observations and theories are very interesting. Generating adversarial examples from the higher layers of the neural network reduces the distance of gradient propagation, and Table 2 also shows the efficiency of the method.

2. The method compares with multiple strong baselines for LLM defense and shows outperformance. The experimental results look good.

3. The paper is clearly presented and easy to follow, with many intuitions discussed in detail and the related work adequately discussed. Although the proposed method has some limitations, it is inspiring for future adversarial training of LLMs.

**Weaknesses:**

1. The three observations mentioned in the abstract appear not to be discussed in detail in the paper. How do these observations motivate the methodology?

2. How effective are SMAAT generated adversarial examples compared to standard adversarial examples in attacking LLMs?

3. Are the baseline and adversarial training models being compared in the paper using the same augmentation or training strengths? The strength and budget of the baselines appear not to be presented in the paper.

4. It is unclear about the generalizability of the method. Will the proposed method be effective against other types of attacks such as typos?

**Questions:**

See the above

---

> ### Author Response · Authors · 2023-11-21
>
> We appreciate your valuable feedback and constructive comments.
>
> > The three observations mentioned in the abstract appear not to be discussed in detail in the paper. How do these observations motivate the methodology?
>
> **Response:** In our revision, we have further clarified how the three fundamental observations motivated our approach. For this, we expanded Section 3.0 to also include our intuitions that drove our problem formulation.
>
> > How effective are SMAAT generated adversarial examples compared to standard adversarial examples in attacking LLMs?
>
> **Response:** AEs generated in the last layer are expected to be more potent compared to AEs generated at the initial layer as attack strength can be more freely adjusted. To test this, we deployed the AEs generated by SMAAT at the last layer by feeding them to the last layer of the BERT model. Our results obtained on three datasets show that the robustness of the standard BERT model reduced to 0 in all cases. This suggests that AEs generated with SMAAT are more effective than those created in the initial layer.
>
> > Are the baseline and adversarial training models being compared in the paper using the same augmentation or training strengths? The strength and budget of the baselines appear not to be presented in the paper.
>
> **Response:** The strength of AEs in each model varies among methods. PGD-based AT methods that rely on initial layer training are subject to low epsilon values. Whereas our approach, which generates AEs in the last layer, is not subject to such constraint. In contrast, for methods like TMD, SAFER, and RSMI the strength of AE cannot be defined in the same manner. Therefore, in our experiments, we adhered to the hyperparameters recommended in their respective original publications. All models were trained for an equivalent number of epochs.
>
> We believe overall training complexity of these methods can best be assessed in terms of their training run-times. In this regard, SMAAT brings only a marginal run-time overhead to the clean models, which took only a couple of second for BERT and RoBERTa.
>
> > It is unclear about the generalizability of the method. Will the proposed method be effective against other types of attacks such as typos?
>
> **Response:** To more comprehensively evaluate the effectiveness of our method against a wider array of attacks, we carried out tests using the datasets included in the AdvGLUE benchmark. This benchmark encompasses AEs generated from 17 distinct attack types, typo based TextBugger included. The results of these tests are shown in Table 3 of the revised manuscript. The performance of our method mirrors the trends observed in our preliminary assessments. SMAAT exhibits average robustness improvements of 5.6% and 2.6% on the two new benchmarks for BERT, and 12.1% and 2.5% for RoBERTa, in comparison to standard and FreeLB++ models, respectively, while maintaining comparable generalization.

---

> > ### Author Response · Authors · 2023-11-23
> >
> > Dear Reviewer  6Kya,
> >
> > Thank you once again for your valuable feedback. We have diligently incorporated your suggestions into the manuscript. We are eager to engage in further discussions and address any additional concerns you may have.
> >
> > Best regards

---

### Official Review · Reviewer_crCo · 2023-11-03

**Soundness:** 2 fair
**Presentation:** 2 fair
**Contribution:** 2 fair
**Rating:** 5
**Confidence:** 4

**Summary:**

This paper proposes an efficient adversarial training (AT) method, particularly for language models, called SMAAT. SMAAT speeds up AT by only using the last several layers to generate adversarial samples. In this way, it does not need to back-propagation through all the layers while generating adversarial training data, thus reducing the training time. Empirical results seem to validate the effectiveness of SMAAT in efficiently learning a robust and generalizable language model.

**Strengths:**

1. The empirical results on three datasets seem to justify the effectiveness of the SMAAT.

2. The authors try to provide some theoretical derivation to motivate the proposed method.

**Weaknesses:**

1. I am confused about the definition of ‘off-manifold’ and ‘on-manifold’. It would be better for the authors to provide more clear definitions and high-level explanations to help understand.

2. I am confused about the choice of $l^* = n$. In the objection function in Eq. (10), it seems that any $l^*$ satisfies when $ID(i-1)<ID(i), \forall I < l^*$. Based on my observation of Figure 3, $l^*$ can be chosen any number smaller than 13 since ID is monotonically decreasing. Therefore, based on Eq. (10), I cannot understand why the choice of $l^* = n$ is optimal.

3. Empirical results are limited. The evaluation should be conducted on various datasets in the GLUE benchmark.

4. The title seems to overclaim the contribution of this paper. All the results in this paper are shown on two language models, i.e., BERT and RoBERTa. However, I did not see the results of ‘large language models (LLMs)’ such as Llama2-70b. I am wondering whether the proposed method can be scalable to LLMs. If so, please show the empirical justification.

Minor comments: keep a consistency of the term ‘ROBERTA’ in Section 3.3 and ‘RoBerta’ in Introduction.

**Questions:**

Please refer to “Weaknesses”.

---

> ### Author Response · Authors · 2023-11-21
>
> We thank the reviewer for their feedback on our work.
>
> > I am confused about the definition of ‘off-manifold’ and ‘on-manifold’. It would be better for the authors to provide more clear definitions and high-level explanations to help understand.
>
> **Response:** The manifold hypothesis stands as one of the most compelling explanations for the susceptibility of deep neural networks to adversarial samples. This hypothesis fundamentally posits that data resides on a low-dimensional manifold within a high-dimensional representation space, and that a network, during training, learns to approximate this manifold. Consequently, an off-manifold sample, deviating from this foundational manifold, leads to undefined behavior in the network. Accordingly, an off-manifold sample is one that diverges from the underlying manifold and the network’s behavior is undefined. For better clarity, we have incorporated this explanation into Section II (last paragraph of page 3) of the revised paper.
>
> > I am confused about the choice of $l^*=n$. In the objection function in Eq. (10), it seems that any $l^*$ satisfies when $ID(i-1) , ID(i), \forall i < l^*$. Based on my observation of Figure 3, $l^*$ can be chosen any number smaller than 13 since ID is monotonically decreasing. Therefore, based on Eq. (10), I cannot understand why the choice of $l^*=n$ is optimal.
>
> **Response:** In our definition, the optimal layer is the one that minimizes the length of the back-propagation path. Therefore, it is preferable to perform adversarial training at output layers as opposed to input layers, which will enhance scalability. The main finding of our research is the demonstration that when the intrinsic dimensionality of the feature manifold decreases monotonically, adversarial training can be effectively performed at the layer with the lowest intrinsic dimension. As evidenced by the data in Figure 4, the final layer—the one with the highest index—exhibits the lowest intrinsic dimension. Therefore, implementing adversarial training at this layer not only optimizes generalization and robustness but also accomplishes these goals with minimal computational cost.
>
> > Empirical results are limited. The evaluation should be conducted on various datasets in the GLUE benchmark.
>
> **Response:** We further evaluated our method on four datasets in the GLUE benchmark (including SST2, QQP , QNLI, RTE). Rather than performing adversarial attacks separately, as in Table 1, we preferred to use the AdvGLUE benchmark which includes the adversarial counterparts of the same for four datasets. The outcomes of these evaluations are presented in Table 3 of the revised paper. Notably, the patterns identified in other benchmarks are consistently observed here as well. Overall, SMAAT exhibits average robustness improvements of 5.6% and 2.6% on the two new benchmarks for BERT, and 12.1% and 2.5% for RoBERTa, in comparison to standard and FreeLB++ models, respectively, while maintaining comparable generalization.
>
> > The title seems to overclaim the contribution of this paper. All the results in this paper are shown on two language models, i.e., BERT and RoBERTa. However, I did not see the results of ‘large language models (LLMs)’ such as Llama2-70b. I am wondering whether the proposed method can be scalable to LLMs. If so, please show the empirical justification.
>
> **Response:** We extended our experiments to include a fine-tuned version of the quantized LLAMA-2 7B model, utilizing PEFT. In this process, we discovered that the intrinsic dimensionality of the representations generated by this model is non-decreasing, mirroring the trend observed in vision models. Additional information about these experiments is presented in Appendix Section H.

---

> > ### Comment · Reviewer_crCo · 2023-11-22
> >
> > Hi Authors,
> >
> > I appreciate the additional experimental results on GLUE, AdvGLUE, and Llama2-7B results. My concerns have been partially resolved. I am willing to increase my score. However, I still have concerns as follows:
> >
> > 1. It is still unknown whether the method is applicable to extremely large-scale LLM (i.e., llama2-70B). Since SMAAT still needs expensive back-propagations, it still requires heavy computational overhead.
> >
> > 2. SMAAT chooses only the last layer for generating adversarial perturbations to achieve computational efficiency. I suppose choosing all the layers to calculate adversarial perturbation can provide a more accurate approximation of the perturbations, thus leading to improved performance. However, this is not the same case as the empirical results. As FreeLB which uses all the layers has a worse performance than SMAAT, I am wondering the explanation for this phenomenon. Besides, is there any ablation study on the layer number chosen for generating adversarial perturbations?

---

> > > ### Author Response · Authors · 2023-11-22
> > >
> > > Thank you for your quick response and reconsideration of your score.
> > >
> > > >It is still unknown whether the method is applicable to extremely large-scale LLM (i.e., llama2-70B). Since SMAAT still needs expensive back-propagations, it still requires heavy computational overhead.
> > >
> > > **Response:** We must emphasize that SMAAT is the most efficient adversarial training approach because it requires only backpropagation through a single layer. For the 7B model, SMAAT requires updating only 131M parameters during the backpropagation step. This number would be expected to double for the 70B model given the availability of GPU resources to load all the parameters of those large models.
> > >
> > > >SMAAT chooses only the last layer for generating adversarial perturbations to achieve computational efficiency. I suppose choosing all the layers to calculate adversarial perturbation can provide a more accurate approximation of the perturbations, thus leading to improved performance. However, this is not the same case as the empirical results. As FreeLB which uses all the layers has a worse performance than SMAAT, I am wondering the explanation for this phenomenon.
> > >
> > > **Response:** We believe the superiority of SMAAT in comparison to FreeLB is due to three reasons:
> > > 1. The last layer of a model includes adversarial examples from all the previous layers when intrinsic dimensionality decreases monotonically. Thus, approaches like FreeLB which perform training at the first layer see fewer adversarial examples.
> > > 2. Performing adversarial training at the last layer allows choosing the PGD perturbation strength more freely as it does not impose any input space perturbation constraint.
> > > 3. Since the last layer has the lowest intrinsic dimension, it may be easier to create off-manifold adversarial examples as compared to layers with higher intrinsic dimensions.
> > >
> > > >Besides, is there any ablation study on the layer number chosen for generating adversarial perturbations?
> > >
> > > **Response:** The ablation analysis on conducting AT in the intermediate is added to Sec 4.2 (Fig. 4, page 8). These results show that the robustness of the model decreases as the trained layer gets close to the input layer. This finding is in line with Theorem 3.1 which posits that when the model has monotonically decreasing intrinsic dimensionality,  moving the AT to deeper layers offers better robustness.

---

### Official Review · Reviewer_Tdyi · 2023-11-03

**Soundness:** 3 good
**Presentation:** 3 good
**Contribution:** 3 good
**Rating:** 8
**Confidence:** 4

**Summary:**

The paper presents SMAAT, an efficient approach for adversarial training of language models, which relies on fine-tuning a pretrained model using adversarial examples generated from the last layer of the model. To motivate the approach, the paper discusses the link between inputs being out of manifold for consecutive layers, and the difference in the intrinsic dimensionality of these layers. Then, since language models exhibit a monotonically decreasing *intrinsic dimensionality (ID)* of their representations throughout layers, the optimal layer to generate adversarial examples from is the last layer. Using only the last layer in the forward-backward passes of PGD to generate adversarial examples is much more efficient than using the full model. Finally, the paper shows that on top of the efficiency of the proposed approach, the defended models are more robust than recent defenses against common adversarial attacks for text on three text classification benchmarks.

**Strengths:**

- **Robustness and Natural Accuracy:** The paper demonstrates substantial improvements in model robustness when compared to other defense methods, without compromising natural accuracy. It achieves consistent results across three benchmark datasets using two different models.
- **Efficient Adversarial Training:** The paper offers a very efficient approach to adversarial training. By concentrating on the last layer of the model for generating adversarial examples, it significantly reduces the computational overhead associated with this process. This efficiency is a significant contribution, making adversarial training more accessible for practical applications.
- **Clear Presentation and Context Setting:** The paper is well-organized and presents related work in the field clearly. The description of the approach is also easy to understand.

**Weaknesses:**

In general, the formulation and description of the motivation behind the approach is not very clear and lacks rigor. This makes the foundations of the approach unsound. There are several quantities vaguely defined, such as the basis $U_l$ obtained from the SVD, the formulation of the theorem 3.1 and its proof, or the Intrinsic Dimension of a layer that is discussed and used before being defined. This leads to critical misunderstandings that require clarification.

### Typos
Here are some typos I noticed:
- conjoncture -> conjecture (in figure 2 and in last paragraph of page 3)
- the $\exists$ should be $\forall$ in equation 4 and equation 9
- orhonormal -> orthonormal (in 3.1, between equation 4 and equation 5)
- Emperical -> Empirical (in title of 3.3)

**Questions:**

- As mentioned in the paper, the basis $U_l$ obtained through SVD is an orthonormal basis. Thus, $U_l U_l^\top = I$, which makes the projection error as defined in the paper always equal to zero.
- Can the authors clarify how to derive equation 7 from equation 6 ? It is critical as it links the search for the optimal layer $l^*$ to theorem 3.1.
- In theorem 3.1, the proof in Appendix is given for the opposite side: If $||(I - U_{(i-1)} U_{(i-1)}^\top)\delta_{(i-1)}|| < ||(I-U_i U_i^\top)\delta_i||$ then $rank(i-1) < rank(i)$. There is also the equality case that is included in the theorem but not in the proof. Maybe the inequalities should be reversed in the formulation of the theorem, which would make a proof by contraposition ? This would also make sense with the remaining of the paper, since we observe decreasing ID and not increasing ID.
- Similarly, for the objectives of SMAAT (eq. 9 and 10), the inequalities on the ID do not make sense since we observe decreasing ID throughout the layers. I think the inequality should be reversed as well, i.e. $ID(i) < ID(i-1)$.
-  What is k and $U_l^k$ in the computation of ID (equation 11) ? I assume it is the k first rows and columns of $U_l$, which would make ID close to the rank of the representation. This should be clarified, as well as the link between ID and the rank, since the theorem is stated using the *rank operator*.
-  What is $\lambda_{max}$ ? Is it the maximum eigenvalue ? If so, I'm unsure about the assumption used in the proof of theorem 3.1 about the ratio of $\lambda_{max}$. There is an additional layer in the network from which the jacobian is computed in the numerator, thus the Lipschitz constant of the numerator is greater than the one of the denominator, making the ratio greater than 1. Does the remaining of the proof holds with this ?
-  The proposed approach is interestingly the opposite of YOPO [1], can the authors develop on the link and differences regarding this method ?

[1] Zhang et al., You Only Propagate Once: Accelerating Adversarial Training via Maximal Principle, NeurIPS 2019.

---

> ### Author Response · Authors · 2023-11-21
>
> Thank you for carefully reading our paper and for the suggestions. We corrected the mathematical representation and revised the proof of Theorem 3.1 accordingly.
>
> > As mentioned in the paper, the basis $U_l$ obtained through SVD is an orthonormal basis. Thus, $U_lU_L^T = I$, which makes the projection error as defined in the paper always equal to zero.
>
> **Response:** We compute the projection error using the eigenvectors corresponding to the top-k eigenvalues, where k represents the ID of the respective layer. Therefore, samples aligned with the data manifold exhibit lower projection errors, while others deviate with higher errors in the k-dimensional data manifold. Our notation is modified to include the index $k$, representing the top-k eigenvectors, which translates into the effective ID of the layer.
>
> >Can the authors clarify how to derive equation 7 from equation 6 ? It is critical as it links the search for the optimal layer $l^*$
>  to theorem 3.1.
>
> **Response:**  To enhance the clarity of the derivation, we introduced an intermediate (Eq. 7). We also note that Eq. 8 (old Eq. 7) sets a sufficient condition to satisfy Eqs. 6 and 7.
>
> > In theorem 3.1, the proof in Appendix is given for the opposite side: If $\|(I-U_{(i-1)}U_{(i-1)}^T)\delta_{(i-1)}\| < \|(I-U_iU_i^T)\delta_(i)\|$ then rank(i-1) < rank(i). There is also the equality case that is included in the theorem but not in the proof. Maybe the inequalities should be reversed in the formulation of the theorem, which would make a proof by contraposition ? This would also make sense with the remaining of the paper, since we observe decreasing ID and not increasing ID.
>
> **Response:** We appreciate your insightful observation. We have carefully revised Theorem 3.1 and its proof. The notation is now coherent and the direction of the inequalities are correct.
>
> >Similarly, for the objectives of SMAAT (eq. 9 and 10), the inequalities on the ID do not make sense since we observe decreasing ID throughout the layers. I think the inequality should be reversed as well, i.e. $ID(i) < ID(i-1)$
>
> **Response:** In accordance with our initial description in the text, the direction of inequalities are corrected.
>
> > What is k and $U_l^k$ in the computation of ID (equation 11) ? I assume it is the k first rows and columns of $U_l$, which would make ID close to the rank of the representation. This should be clarified, as well as the link between ID and the rank, since the theorem is stated using the rank operator.
>
> **Response:** This is correct: the variable $k$ represents the number of top eigenvalues needed to obtain a projection similar to the original samples and also used as the ID of features. We revised the paper for better clarity.
>
> > What is $\lambda_{max}$? Is it the maximum eigenvalue? If so, I'm unsure about the assumption used in the proof of theorem 3.1 about the ratio of $\lambda_{max}$. There is an additional layer in the network from which the jacobian is computed in the numerator, thus the Lipschitz constant of the numerator is greater than the one of the denominator, making the ratio greater than 1. Does the remaining of the proof holds with this?
>
> **Response:** We have updated the mathematical formulation of the theorem to be consistent with the revised assumptions. In the new version of the theorem, we employed a proof by contraposition to establish its validity. We believe these modifications greatly contribute to the clarity and accuracy of our work.
>
> > The proposed approach is interestingly the opposite of YOPO [1], can the authors develop on the link and differences regarding this method ?
>
> **Response:** We believe this seeming contradiction is due to differences between vision and text domains. In Section G (Figure 6) of the revised paper, we present the measured intrinsic dimensionality of the ViT model which is noticeably different from that of text models. In fact, our approach suggests that the optimal layer $l^*$ for the ViT model would be the initial layer due to the increasing ID.  We have incorporated this clarification into the paper to provide a more nuanced understanding of the relationship between layer-wise behavior and robustness.
>
> > Typos
>
> **Response:**  We corrected typos. However, in Eqs. 4 and 10 (old Eq. 9), we define an AE to be off-manifold in any layer, and not in all of the layers. Therefore,  we kept $\exists$ notation instead of $\forall$.

---

> > ### Comment · Reviewer_Tdyi · 2023-11-22
> >
> > Thank you for the detailed answer, the revision of the paper and all additional work.
> > - The revision of the proof and the derived results seems correct to me now. The results look more solid now and all my concerns have been addressed.
> > - I appreciate the discussion around ViT and LLAMA-2 in the Appendix, it nicely connects this work to YOPO for vision tasks, and gives a better understanding of the inner behaviors. However, I think including the obtained results as a table, even though they are negative, would be better than only stating that the results are not improved in the associated text. Otherwise, this information might be missed by the reader.
> > - I appreciate the additional results on GLUE, AdvGLUE, using a stronger attack, and the comparative study of AT at different layers.
> >
> > I like the idea of the paper and find the developments interesting. I'm increasing my score to an 8 (accept)

---

> > > ### Author Response · Authors · 2023-11-23
> > >
> > > We sincerely appreciate the reviewer's highly positive comments. For better clarification, we have included our results in Table 4 in Appendix Section E.

---

### Official Review · Reviewer_JQp6 · 2023-11-06

**Soundness:** 3 good
**Presentation:** 3 good
**Contribution:** 2 fair
**Rating:** 6
**Confidence:** 3

**Summary:**

The paper introduces SMAAT, an efficient Adversarial Training method that uses only adversarial examples generated in the last layer of a model in encoder-based large language models. The proposed method has fast training and inference speed since we do not have to do a full forward-backward passes.

**Strengths:**

- The proposed method is intuitive and backed by good motivation, and both theoretical and experimental findings.
- The proposed method outperforms most of the previous methods on various attacks and datasets.

**Weaknesses:**

- Argument about the manifolds between the layers is not very clear to the reader.
- Results on Table 1 are difficult to interpret. I would suggest just to boldify the best result for every attack in each dataset without having any underlined results.
- I believe that the method is probably not very novel or of high contribution.

**Questions:**

- Have you conducted any experiments where you use adversarial examples generated from other layers instead of the only the last one. Sometimes theorems and experiments can provide very different outcomes.
- I read the limitations at the end of the paper. Although this is not in the scope of this work I would suggest to still try this method with image data.

---

> ### Author Response · Authors · 2023-11-21
>
> Thank you for your time and suggestions.
>
> > Argument about the manifolds between the layers is not very clear to the reader.
>
> **Response:**  We revised our statement in Section I (next to last paragraph on page 2) to better clarify the manifold relation between layers.
>
> > Results on Table 1 are difficult to interpret. I would suggest just to boldify the best result for every attack in each dataset without having any underlined results.
>
> **Response:**  The best results in Table 1 are in bold.
>
> > I believe that the method is probably not very novel or of high contribution.
>
> **Response:**  We would like to emphasize that the novelty of our method lies in (i)  the discovery that the intrinsic dimensionality of internal representations in fine-tuned decoder models monotonically decreases and that, in this case, (ii) applying AT in the higher layers covers all the AEs from the previous layers and offers better robustness than applying AT in the initial layers. This justifies (iii) applying PGD-based adversarial training to only to the last layer. Our method not only improves robustness and generalization capabilities but also achieves these improvements at a significantly lower computational cost compared to state-of-the-art methods. In our revision, we have extended our experimental analysis to include more tasks/datasets, thereby reinforcing the efficacy of our method.
>
> > Have you conducted any experiments where you use adversarial examples generated from other layers instead of the only the last one. Sometimes theorems and experiments can provide very different outcomes.
>
> **Response:**  As part of our revision, we conducted additional experiments focusing on the application of adversarial training to intermediate layers. The outcomes of these experiments are presented in Figure 4 of the revised manuscript. In agreement with our hypothesis, the results demonstrate a reduction in robustness when the adversarially trained layer is closer to the input layer, and the best robustness is achieved in the last layer.
>
> > I read the limitations at the end of the paper. Although this is not in the scope of this work I would suggest to still try this method with image data.
>
> **Response:**  We extended the application of our method to (i) vision and (ii) LLM domains. To achieve this, we evaluated the intrinsic dimensionality of the ViT  model and the quantized LLAMA-2 7B model. Comprehensive details about these models and the outcomes of these experiments are presented in Appendices G and H. Our findings reveal that both models have non-monotonic variations in intrinsic dimensionality. We further applied SMAAT to both models to test our hypothesis. We did not observe enhanced robustness, as expected by our formulation.

---

> > ### Comment · Reviewer_JQp6 · 2023-11-21
> >
> > Thank you for the fast and detailed reply. I do like the idea and the results look competitive even though there is an incremental improvement in the adversarial robustness, and in some cases the proposed method does not outperform the SOTA methods. I have also skimmed through the new edits in the paper and they look good. I would prefer for the ViT and LLM experiments to be moved in the main text instead of the end of the Supplement, and be more extensive. For instance you could try to run similar experiments to the ones that you have for BERT and RoBERTa in the main text. I will raise my score to 6, but I will keep an eye for the rest of the reviewers' opinions.

---

> > > ### Author Response · Authors · 2023-11-22
> > >
> > > We appreciate your fast response and reconsideration of your score for our work.
> > >
> > > > For instance you could try to run similar experiments to the ones that you have for BERT and RoBERTa in the main text.
> > >
> > > **Response:** Implementing all the attacks in Table 1 on the LLAMA-2 model requires changing the design of most of those techniques. More specifically,  the certified robustness-based approaches, such as RSMI and SAFER, rely on random masking of input tokens which does not directly apply to causal language models like LLAMA-2. Additionally, for the data augmentation-based approaches like ASCC and DNE, the computational cost would be prohibitively high.  Therefore, we could only extend our experiments for the   FreeLB++ method in addition to SMAAT.  Our measurements revealed that both methods yielded 94% and 71% accuracy on SST-2 and its adversarial version, which was the result obtained on the standard (vanilla) method. This shows that under this setting neither method offers a robustness advantage over the standard.
> > >
> > > > I would prefer for the ViT and LLM experiments to be moved in the main text instead of the end of the Supplement
> > >
> > > **Response:**  We tried this. However, due to the page limitation,  this requires moving some of the other important findings of the work to the supplementary. In our evaluation, any such modification will hinder the readability of the text. To increase the visibility of these results, however,  we moved them to an earlier section, to Sec. D and E from Sec. G and H,  of the supplementary.

---

### Author Response · Authors · 2023-11-21

We extend our gratitude to the reviewers for their insightful feedback. We have updated our paper submission to address reviewer comments so far while our revisions in the updated paper are marked in **blue-colored** text. We provide a summary of key improvements to the paper we have made based on reviewer comments, and we welcome additional feedback and remain open to further suggestions and questions from reviewers. Individual responses follow.

**Ablation study:** Our intuition behind SMAAT is: if the model has a monotonically decreasing intrinsic dimension (ID), off-manifold AEs cumulatively increase when moved to the higher layers. Therefore applying AT in the last layer of these models offers better robustness. To validate our approach we applied AT in the middle layers, and as in line with our hypothesis robustness increases when we move to higher layers. Please see Figure 4 in the updated submission.

**Adaptive attacks:** We design adaptive attacks to approximate the lower bound of empirical robustness of SMAAT. Specifically, we increase the PGD steps to 50 and we change the attack type to target. Also, since SMAAT robustifies the last layer, we applied “feature adversaries” that create adversarial examples by targeting the model’s internal representation. Although this leads to a significant reduction in model robustness, SMAAT consistently outperforms other models.

**More dataset and attack:** We evaluate SMAAT on GLUE and advGLUE benchmarks to better assess its effectiveness. Note that advGLUE contains 17 different attacks including typo attacks.

**Formulation, typo fixing and clarification:** Thank you to reviewers that pointed out inequality mismatches and typos. We fixed them. We also clarified the raised ambiguities in Theorem 3.1.

**Application to vision and generative models:**  We applied SMAAT to vision and generative models. However, all baselines from these two families of models show similar non-decreasing ID  behavior. So, these models do not satisfy the constraints in SMAAT objective (Eq. 10). Yet, as a sanity check, we applied SMAAT on these models, as expected, we didn’t obtain an improvement in robustness..

---

### Meta-Review · Area_Chair_x3Ks · 2023-12-04

**Metareview:**

In this paper, authors are concerned with the definition of more efficient adversarial training approaches. That's a quite relevant research topic from a practical perspective, especially considering the trend in the community towards the use of ever larger models. This paper claims and shows supporting evidence that gradient-based attacks typically used for adversarial training can be computed using only a subset of the model's layers, which is useful not only to reduce the overhead adversarial training incurs, but also to enable adversarial training for models that operate on discrete domains such as models of text for instance.

As highlighted by the reviewers and on the above, the paper is interesting and tackles a very relevant problem. The proposed solution is simple and efficient and has the potential to drastically reduce the overhead adversarial training incurs. However, the empirical evaluation's scope is a bit too narrow, and the extent to which the proposal actually works is somewhat unclear. The evaluation focused on text classification tasks, and only showed confirming evidence for a small set of models. Some preliminary experiments were carried out on other situations during the discussion period, and those do not strongly support broad usefulness. It's also unclear to what extent generating adversarial perturbations at the representation domain is a sensible approach for models of a discrete domain. It could be the case that perturbing the continuous representation space yields representations of unnatural trivial adversaries.

I would recommend expanding the evaluation significantly prior to publication.

**Justification For Why Not Higher Score:**

The paper is quite interesting and potentially impactful. However, the empirical assessment is too limited to the extent that its usefulness is not clear.

**Justification For Why Not Lower Score:**

N/A

---

### Decision · Program_Chairs · 2024-01-16

Reject